# StarGraph:
# Knowledge Representation Learning based on Incomplete Two-hop Subgraph

## Abstract

Conventional representation learning algorithms for knowledge graphs (KG) map each entity to a unique embedding vector, ignoring the rich information contained in the neighborhood. We propose a method named StarGraph, which gives a novel way to utilize the neighborhood information for large-scale knowledge graphs to obtain entity representations. An incomplete two-hop neighborhood subgraph for each target node is at first generated, then processed by a modified self-attention network to obtain the entity representation, which is used to replace the entity embedding in conventional methods. We achieved SOTA performance on ogbl-wikikg2 and got competitive results on fb15k-237. The experimental results proves that StarGraph is efficient in parameters, and the improvement made on ogbl-wikikg2 demonstrates its great effectiveness of representation learning on large-scale knowledge graphs.

## 1 Introduction

A Knowledge Graph (KG) is a directed graph with real-world entities as nodes and relationships between entities as edges. In this graph, each directed edge together with its head and tail entities forms a triple (head entity, relation, tail entity), indicating that the head and tail entities are connected by a relation.

Knowledge graph embedding (KGE), also known as knowledge representation learning (KRL), aims to embed entities and relations into low-dimensional continuous vector spaces to characterize their latent semantic features. A scoring function is defined to measure the plausibility for triples in such spaces, then the embeddings of entities and relations are learned by maximizing the total plausibility of the observed triples. These learned embeddings can be used to implement various tasks such as knowledge graph completion (Bordes et al., 2013; Wang et al., 2014), relationship extraction (Riedel et al., 2013), entity classification (Nickel et al., 2011), etc. The plausibility of each triple is calculated on the embeddings of the entities and relations in it, and the embeddings are directly taken out from the embedding tables. Such a shallow lookup decides that those models are inherently transductive. Moreover, the rich contextual information contained in the neighboring triples is not taken into account.

Compared with shallow embedding models, methods that are able to encode neighborhood information, usually perform much better across various KG datasets (Zhang & Chen, 2018; Zhang et al., 2021; Wang et al., 2019). Any generic graph neural networks could be employed as the encoder. However, there is a problem adopting these methods to large-scale knowledge graphs, for previous work (Nathani et al., 2019; Wang et al., 2020) takes the multi-hop subgraph of the node as input. Due to the large number of nodes and edges, multi-hop subgraphs in large-scale graphs can easily exceed the size limitation, and the subgraphs generation and network calculations can both be very time-consuming.

The neighborhood surely contains information for the target node, therefore can be used for learning its representation. In order to adopt neighborhood neural encoders in large-scale KG, an intuitive idea is to utilize partial neighborhood information instead of the complete multi-hop subgraph.

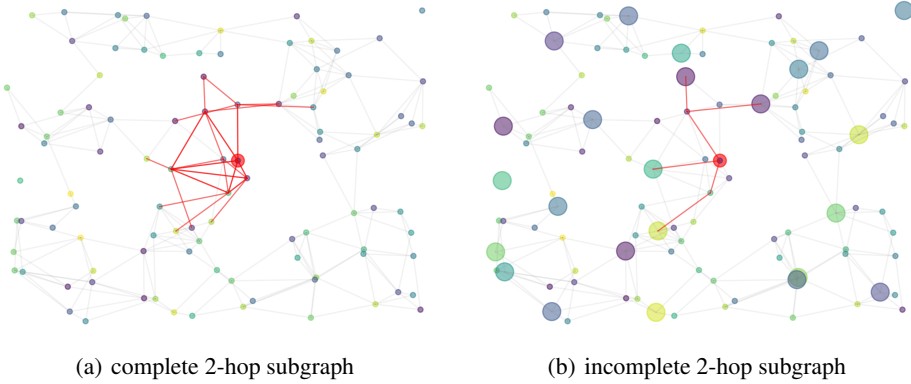

(a) complete 2-hop subgraph                    (b) incomplete 2-hop subgraph

Figure 1: Illustration of a subgraph generated by StarGraph. Dots and lines represent nodes and edges in the graph, respectively, with larger dots indicating anchors. The color red indicates the example target node and the sampled subgraph.

In this paper, we propose to learn the knowledge representation for each target node based on its incomplete 2-hop neighborhood subgraph. Neighbors out of reach within two hops are not as closely related to the target node according to us, so are not taken into consideration. An example of a complete 2-hop subgraph is given in Figure 1(a), where we can see, even in such a small knowledge graph, the 2-hop subgraph comprises quite a few nodes and edges and seems to contain a lot of redundant information. It is more efficient to construct a proper incomplete subgraph with a few nodes and edges.

We got inspiration from the anchor-based strategy (Galkin et al., 2022), which selects a small fraction of all the graph nodes as anchors and learn embeddings only for anchors instead of all nodes. In our work, we sample anchors from the 2-hop neighborhood, along with the edges to reach each anchor, to construct the incomplete 2-hop subgraph, which is illustrated in Figure 1(b). The incomplete subgraph is not restricted to contain only anchors, we can also sample some general nodes into the subgraph as supplementary information.

In order to reasonably model the subgraph structure and enable sufficient interaction of node embeddings, we adopt a self-attention network to extract the neighborhood information. Taking the characteristics of knowledge graphs into consideration, we modify the attention module to be more efficient and propose a novel way to embed the edges in a graph.

Comparing the subgraphs in Figure 1(a) and (b), the trimmed subgraph is much more efficient and is likely to be more effective to describe the target node, especially for large-scale knowledge graphs. And this is demonstrated by our experimental results. The way a node being represented by the incomplete subgraph is like how to locate a star in the sky, which is to use a few bright stars (anchors) to indicate the location. For the incomplete subgraph in the entire graph looks like a constellation among the stars, we call the proposed method *StarGraph*.

## 2 RELATED WORK

**Distance-based models** consist a major branch of knowledge graph embedding methods. TransE (Bordes et al., 2013) embeds relations and entities into the same vector space, and the relation embedding is interpreted as a translation from the embeddings of the head entity to the tail entity. RotatE (Sun et al., 2018) uses the rotations of vectors to explain various relations, where the inverse relations can be modeled by complex embeddings. PairRE (Chao et al., 2021) propose to learn two embeddings for each relation, respectively used to map head and tail entity embeddings into the corresponding relation space. TripleRE (Yu et al., 2021) learns another embedding for each relation on the basis of PairRE, and the extra relation embedding is used as a translation vector between the mapped entity embeddings of head and tail.

**Contextual information** has been widely used for knowledge graphs to enhance model performance, but in many different ways. CoKE (Wang et al., 2019) treats link prediction as a natural language processing (NLP) task, and trains the NLP model with long entity-relation-sequences taken from the graph, linking entities originally distant from each other more closely. GC-OTE (Tang et al., 2020) directly integrates the graph context into the distance scoring function by considering tail/head entities with the same head/tail entity and relation all at once and minimizing the distance between their embeddings. NodePiece (Galkin et al., 2022) borrows the idea of subword-tokenization from NLP and proposes to use a small number of nodes as anchors, constructs vocabulary for each entity using surrounded anchors and relations, which implicitly includes contextual information, and calculates node representations from this.

**Graph encoders** can be conventional graph neural networks, e.g, GCN (Kipf & Welling, 2017) and GraphSAGE (Hamilton et al., 2017), or the recently proposed Graph Transformer (Dwivedi & Bresson, 2021) and Graphormer (Ying et al., 2021), whose work are developed on the basis of Transformer (Vaswani et al., 2017). The basic unit in Transformer is the self-attention module, which provides a novel way to deal with the inputs. It takes a sequence of feature vectors, (tokens as being called) as inputs and updates each token with all others within the sequence, which is controlled with an attention matrix. It provides an effective way for information exchange and allows the tokens staying unordered, which seems suitable to process the structure of graphs.

The graph attention networks (Nathani et al., 2019) and graph attenuated attention networks (Wang et al., 2020) base their work on ConvVB (Dai Quoc Nguyen et al., 2018) and CapsE (Vu et al., 2019), which are proved invalid by Sun et al. (2020), thus will not be compared with in this paper.

## 3 METHODOLOGY

The StarGraph is a method to obtain entity representations based on incomplete neighborhood information, which includes two stages, generating and encoding a subgraph for each node. The obtained entity representations, together with the relation embeddings, are used to compute the scores of the triples by a distance-based score function, and optimized with the self-adverse negative sampling loss (Sun et al., 2018). We will describe each part in detail below.

### 3.1 SUBGRAPH GENERATION

To generate the proper incomplete subgraphs, we should start with select part of the nodes to form the anchor set. For ease of description, we denote the anchor set as $A$, all nodes of the total graph as $N$. There are $A \in N$, and $|A| \leq |N|$. Then for each target node, we pick part of the anchors and nodes within its 2-hop neighborhood to construct the subgraph.

### 3.1.1 ANCHOR SET

At first, a minority of nodes are selected as anchors. When the number of anchors is fixed and much less than the number of graph nodes, the most intuitive idea is that the anchors should be selected according to the degree of centrality in order to provide common information to as many nodes as possible. Following this idea, Galkin et al. (2022) employs the deterministic anchor selection strategy where 40% of the total number of anchors are nodes with top Personalized Page Rank (Page et al., 1998) scores, 40% are top degree nodes, but remaining 20% are selected randomly for they found random anchor selection to be as effective as centrality-based strategies. We argue that the random part introduce the uncertainty, and anchor sets generated by the same strategy may have different performances in experiments. Besides, there is no need to use two centrality measures, and we only use the degree of nodes as the measure.

If the selected anchors are too concentrated, it may result in that some nodes are surrounded by too many anchors while some other nodes have no anchors around them. To alleviate this problem, we introduce a hyper-parameter *skip-threshold*. When the percentage of anchors in the neighboring nodes of a node exceeds the threshold, that node will not be selected as an anchor. We also need to specify the *anchor-set-size*, i.e., how many nodes are selected as anchors. The process of generating the anchor set can be seen in Algorithm 1. Detailed discussion on the effect of *skip-threshold* can be found in Appendix A, but in the main text the value is fixed at 0.5.

---

**Algorithm 1:** Generate the anchor set for the graph

---

**Input:** graph_nodes $N$, skip_threshold, anchorset_size
**Output:** anchor_set $A$

1 $sorted\_nodes \leftarrow sortInDescendingOrderOfDegree(graph\_nodes)$
2 $anchor\_set \leftarrow \emptyset$
3 **for** $node \in sorted\_nodes$ **do**
4     $nbors \leftarrow oneHopNeighbors(node)$
5     $anchor\_nbors \leftarrow nbors \cap anchor\_set$
6     **if** $size(anchor\_nbors)/size(nbors) > skip\_thresh$ **then**
7        | **continue**          `// many anchors around, skip this node`
8     **end**
9     $anchor\_set \leftarrow anchor\_set \cup \{node\}$
10     **if** $size(anchor\_set) \geq anchorset\_size$ **then**
11        | **break**          `// got enough anchors, end the loop`
12     **end**
13 **end**

---

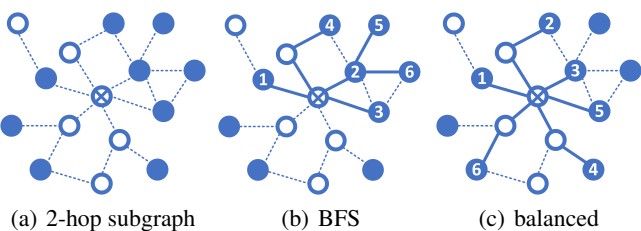

(a) 2-hop subgraph       (b) BFS       (c) balanced

Figure 2: Six anchors sampled from the 2-hop subgraph of the target node $\otimes$ adopting different strategies. Dots and lines represent nodes and edges, where colored dots are anchors and solid lines indicate how we reach the anchors. The numbers on dots indicate the sampling order of anchors.

### 3.1.2 ANCHORS SAMPLING

For each node, a fixed number of anchors are sampled from its 2-hop neighbors, together with the paths to reach the anchors, to construct the incomplete subgraph. When the anchors within two hops are more than needed, for example, sampling six anchors for the target node from its 2-hop subgraph given in Figure 2(a), only the first part of the anchors sampled will be retained. The easiest sampling order to think of is breadth-first-search (*BFS*), but it might induce severe imbalance in anchors distribution, as shown in Figure 2(b). The sampled anchors are concentrated around node#2, and this sampling strategy fails to associate the target node with some non-anchor neighbors.

We propose a more *balanced* sampling strategy that aims to share more anchors with the non-anchor neighbors, by sampling one-hop anchors of the target node and its every non-anchor neighbor in turn. The detailed steps are described in Algorithm 2, and the sampling results of this strategy are shown in Figure 2(c). Note that anchors with smaller degrees will be selected in preference according to line#7 and #13 in Algorithm 2. It is based on the considering that 1) less-popular anchors make each target node more recognizable from others; 2) less-popular anchors can be visited more often to alleviate the long-tail distribution problem in the usage of anchors.

### 3.1.3 NEIGHBORS & CENTER

*Neighbors* refer to a fixed number of nodes sampled from one-hop neighborhood of the target node, and *center* refers to the target node itself. We use these nodes as additional information to further enrich the description of the target node. Different from anchors, that are picked from the predefined anchor set, neighbors and centers could be any node in the graph. We sample the neighbors in a degree-decreasing order, trying to use them to make a common description for each target node.

---

**Algorithm 2:** Sample anchors for subgraph of each node

---

**Input:** anchor_set $A$, node, sample_size
**Output:** subgraph_anchors
1 **Def** $oneHopAnchors(node)$ **as** $oneHopNeighbors(node) \cap anchor\_set$
2 $non\_anc\_nbors \leftarrow oneHopNeighbors(node) - oneHopAnchors(node)$
3 $subgraph\_anchors \leftarrow \emptyset$
4 **while** $size(subgraph\_anchors) < sample\_size$ **do**
5    $remaining\_ancs \leftarrow oneHopAnchors(node) - subgraph\_anchors$
6    **if** $remaining\_ancs \neq \emptyset$ **then**
7       $next\_anc \leftarrow nodeWithMinimumDegree(remaining\_ancs)$
8       $subgraph\_anchors \leftarrow subgraph\_anchors \cup \{next\_anc\}$
9    **end**
10    **for** $nbor \in non\_anc\_nbors$ **do**
11       $remaining\_ancs \leftarrow oneHopAnchors(nbor) - subgraph\_anchors$
12       **if** $remaining\_ancs \neq \emptyset$ **then**
13          $next\_anc \leftarrow nodeWithMinimumDegree(remaining\_ancs)$
14          $subgraph\_anchors \leftarrow subgraph\_anchors \cup \{next\_anc\}$
15       **end**
16    **end**
17 **end**

---

## 3.2 SUBGRAPH ENCODING

A self-attention network (Vaswani et al., 2017), whose input is considered as a sequence though, its operation on each element within the sequence is naturally global and undifferentiated. Therefore, we consider it a perfect choice to encode graphs. The network takes the embeddings of nodes within a generated subgraph as inputs, each node treated as a token. Since a self-attention layer has a global receptive field, one layer can adequately mix the information of all nodes. We use a single transformer block, i.e. one attention layer followed by two linear layers, to construct the network. After this, the output node features are aggregated by mean-pooling across the subgraph to get the representation of target node. More discussion on the choice of encoders can be found in Appendix B.

### 3.2.1 REDUCED ATTENTION MODULE

The attention module consumes a lot of computational resources, so we try to explore a more efficient way to implement the attention mechanism. For a standard $k$-heads attention layer, it takes $\mathbf{X} \in R^{L \times D_{hidden}}$ as input, sets the dimension of each head as $D_{head} = D_{hidden}/k$, and gets the output $\mathbf{Y} = [\mathbf{Y}_1, \mathbf{Y}_2, \cdots, \mathbf{Y}_k] \in R^{L \times D_{hidden}}$ by concatenating the outputs of $k$ heads together. The output of the $i$-th head $\mathbf{Y}_i \in R^{L \times D_{head}}$ is calculated by

$$\mathbf{Y}_i = softmax(\frac{\mathbf{X}\mathbf{W}_{Q,i}\mathbf{W}_{K,i}^T\mathbf{X}^T}{\sqrt{D_{head}}})\mathbf{X}\mathbf{W}_{V,i}, \tag{1}$$

where $\mathbf{W}_{Q,i}, \mathbf{W}_{K,i}, \mathbf{W}_{V,i} \in R^{D_{hidden} \times D_{head}}$ are model parameters to be learned.

The role of the attention module is to integrate information from all nodes in the subgraph. Ensuring this, we make two modifications to attention-head to reduce the amount of computation. Firstly, to get a usable attention matrix, the second dimension of $\mathbf{W}_{Q,i}$ and $\mathbf{W}_{K,i}$ is not forced to equals $D_{head}$, so setting a smaller dimension $D_{attn} \leq D_{head}$ will reduce the computational complexity. Secondly, with an MLP afterward, there is no need to transform the features in the information mixing step. Since $D_{head} = D_{hidden}/k$, we can directly slice $\mathbf{X}$ into $k$ parts with the same shape of $\mathbf{X}\mathbf{W}_{V,i}$ to omit the multiplication operation of $\mathbf{X}$ and $\mathbf{W}_{V,i}$. The output of the $i$-th head $\mathbf{Y}_i \in R^{L \times D_{head}}$ of a reduced attention layer is calculated by

$$\mathbf{Y}_i = softmax(\frac{\mathbf{X}\mathbf{W}_{Q,i}\mathbf{W}_{K,i}^T\mathbf{X}^T}{\sqrt{D_{attn}}})\mathbf{X}_i, \tag{2}$$

where $\mathbf{X}_i \in R^{L \times D_{head}}$ are directly taken out from $\mathbf{X} = [\mathbf{X}_1, \mathbf{X}_2, \cdots, \mathbf{X}_k]$, and $\mathbf{W}_{Q,i}, \mathbf{W}_{K,i} \in R^{D_{hidden} \times D_{attn}}$ are model parameters to be learned.

Beside the attention layer, the computational complexity of MLP can be reduced by decreasing the intermediate dimension $D_{hidden} * mlp\_ratio$. The default value of hyper-parameter *mlp-ratio* is 4, and a smaller value can be considered.

### 3.2.2 PATH INFORMATION FUSION

Treating each anchor as a token, the anchor path, i.e., edges the anchor needs to pass to reach the target node, should be considered as the position information. Instead of following the practice of previous work, where the position embeddings are directly added onto the token embeddings, we propose to take the characteristics of KGE into consideration. Inspired by the score function of TripleRE (Yu et al., 2021),

$$f_r(h, t) = -||\mathbf{h} \circ (\mathbf{r}^h + \mathbf{1}) - \mathbf{t} \circ (\mathbf{r}^t + \mathbf{1}) + \mathbf{r}||, \tag{3}$$

based on the assumption that

$$\mathbf{t} \circ (\mathbf{r}^t + \mathbf{1}) \approx \mathbf{h} \circ (\mathbf{r}^h + \mathbf{1}) + \mathbf{r}, \tag{4}$$

we simplify the conditions and assume that

$$\mathbf{t} \approx \mathbf{h} \circ (\mathbf{r}^h + \mathbf{1}) + \mathbf{r}. \tag{5}$$

Substituting "head, relation, tail" with "anchor, path, token", respectively, we can get token embedding by adding path information to 1-hop or 2-hop anchor as

$$\mathbf{token} = \begin{cases} \mathbf{anchor} \circ (\mathbf{p}_1^a + \mathbf{1}) + \mathbf{p}_1^b & \text{when } path = \{p_1\}, \\ [\mathbf{anchor} \circ (\mathbf{p}_1^a + \mathbf{1}) + \mathbf{p}_1^b] \circ (\mathbf{p}_2^a + \mathbf{1}) + \mathbf{p}_2^b & \text{when } path = \{p_1, p_2\}. \end{cases} \tag{6}$$

Note that after incorporating the path information, the token embedding is more like the representation of the target node rather than the original anchor, therefore mean-pooling is a proper way to aggregate information from the tokens.

### 3.2.3 SUPPLEMENTARY NODES

As described in section 3.1.3, a subgraph may contain nodes sampled from $N$, providing additional information for the target node. As with the entire graph, each node keeps some unique information, while only a small group of nodes are anchors, carrying much general information. Denoting the embedding sizes of $A$ and $N$ as $D_A$ and $D_N$, we set a large $D_A$ to improve the information capacity, and a small $D_N$ to reduce the probability of overfitting. To get the tokens of the same embedding size, $D_A$ is used as $D_{hidden}$, the hidden size for attention module, and the node embeddings are mapped to dimension $D_A$ by a linear layer ahead.

Type embedding is added to the token embedding before the attention module, indicating each node within the subgraph to be the type of anchor, neighbor or center.

### 3.3 SCORE FUNCTION

We adopt a score function that is modified upon TripleRE (Yu et al., 2021), a recently proposed distance-based method. The formula of TripleREv2 is

$$f_r(h, t) = -||\mathbf{h} \circ (\mathbf{r}^h + u \cdot \mathbf{1}) - \mathbf{t} \circ (\mathbf{r}^t + u \cdot \mathbf{1}) + \mathbf{r}||, \tag{7}$$

where $\circ$ denotes the element-wise product, $\mathbf{h}$ and $\mathbf{t}$ correspond to the head and tail entity embeddings respectively, $[\mathbf{r}^h, \mathbf{r}^t, \mathbf{r}]$ comprise the relation embedding, $u$ is a constant specified by users, set to 1 in the original work. We modify equation (7) by switching $\mathbf{1}$ with $\mathbf{r}^h/\mathbf{r}^t$, so that the coefficient $u$ acts on the relation embedding. Now the score of triplets $(h, r, t)$ is calculated as

$$f_r(h, t) = -||\mathbf{h} \circ (u \cdot \mathbf{r}^h + \mathbf{1}) - \mathbf{t} \circ (u \cdot \mathbf{r}^t + \mathbf{1}) + \mathbf{r}|| = -||\mathbf{h} - \mathbf{t} + \mathbf{r} + u \cdot (\mathbf{h} \circ \mathbf{r}^h - \mathbf{t} \circ \mathbf{r}^t)||, \tag{8}$$

which can be interpreted as a boosting method based on TransE and PairRE. The new form is referred to as *TripleRE'* in the rest of this paper. Although TripleRE' is exactly the same as TripleREv2 when $u$ equals 1, according to our experimental results, TripleRE' usually performs better than TripleREv2 during tuning the hyper-parameter $u$.

### 3.4 OPTIMIZATION

We utilize the self-adversarial negative sampling loss (Sun et al., 2018) as objective for training,

$$L = -\log \sigma(\gamma - f_r(h,t)) - \sum_{i=1}^{n} w_i \log \sigma(f_r(h_i, t_i) - \gamma), \quad w_i = \frac{\exp(\alpha f_r(h_i, t_i))}{\sum_{j=1}^{n} \exp(\alpha f_r(h_j, t_j))}, \quad (9)$$

where $\gamma$ is a fixed margin and $\sigma$ is the sigmoid function. We randomly sample $n$ negative triples for each positive one, and $(h_i, r, t_i)$ is the i-th negative triple.

## 4 EXPERIMENTS

### 4.1 DATASETS AND METRIC

**ogbl-wikikg2** (Hu et al., 2020) is a large-scale knowledge graph extracted from the Wikidata knowledge base (Vrandečić & Krötzsch, 2014), containing 2,500,604 entities, 535 relation types and 17,137,181 triplet edges (head, relation, tail). **fb15k-237** (Toutanova & Chen, 2015) is a subset of *fb15k* (Bordes et al., 2013), that consists of triples from Freebase, with inverse relations removed. It contains 14,505 entities, 237 relation types and 310,079 triplet edges.

**Task and Metric**   The task is to predict new triplet edges given the training edges. The evaluation metric follows the standard filtered metric widely used in KG. Specifically, each test triplet edges are corrupted by replacing its head or tail with negative entities. The goal is to rank the true head (or tail) entities higher than the negative entities, which is measured by Mean Reciprocal Rank (MRR).

### 4.2 DETAILED SETTINGS

We set batch-size to 512, negative-sampling-size to 64, and train the model for 500,000 steps on ogbi-wikikg2 and 100,000 steps on fb15k-237. The learning rate is initialized to 0.0001 unless otherwise stated, and decreased by 0.1 at half the maximum training step. Dropout is adopted after each linear layer, and the drop-ratio is 0.05. AdamW is used as the optimizer in our experiments.

For the generated incomplete subgraphs, anchors are always required. The paths and the supplementary nodes are optional, which are not included in subgraphs unless being specified. The number of anchors and neighbors (if used) sampled for each subgraph is fixed to 20 and 5 respectively, and any shortfall is filled with the [PAD] token. The default values for the embedding dimension are $D_A = 256$, $D_N = 32$. Hyper-parameter $u$ in equation (8) is tuned and set to 0.1 on ogbi-wikikg2 and 1.0 on fb15k-237, and $\gamma$ in equation (9) is set to 6.0 following PairRE (Chao et al., 2021).

### 4.3 ABLATION STUDY

**Anchors**   The strategies for anchor set generation and anchors sampling are proposed in section 3.1.1 and 3.1.2 respectively, and display the effects of them in Table 1. Comparison between the first two rows validates the effectiveness of the proposed sampling strategy by a 1.5% gap. The results of the second and third rows are close to each other, despite the uncertainty in *degree+PPR+rand* does not exist in our generation strategy. Comparing the last two rows, the performance gain shows that a larger anchor set performs better, at least under the condition that $|A| \ll |N|$. For the following experiments, the row marked in gray is used as the default setting with anchors unless otherwise specified.

**Network**   The first two rows in Table 2 show that, the larger dimension of features (also known as $D_A$ and *hidden-dim* in this paper) the better result, meantime the higher computational complexity. Comparing the overall performance of network 256-32-4 with its reduced counterpart 256-R32-4, we can see that the reduced attention module proposed in section 3.2.1 saves a little training time without degrading the results. With further adaptation of *attn-dim* and *mlp-ratio*, network 512-R8-2 achieves a better trade-off between the results and the consumption of training time. The row marked in gray is used as the default network for the following experiments, unless otherwise specified.

Table 1: The results on ogbl-wikikg2 with different anchor sets and anchor-sampling strategies.

| Anchor Set | $|A|$ | Sampling | Test MRR | Valid MRR |
|---|---|---|---|---|
| degree+PPR+rand (Galkin et al., 2022) | 20,000 | BFS | 0.6899 | 0.6953 |
| degree+PPR+rand (Galkin et al., 2022) | 20,000 | balanced | 0.7040 | 0.7104 |
| degree-skip0.5 | 20,000 | balanced | 0.7042 | 0.7093 |
| degree-skip0.5 | 80,000 | balanced | 0.7084 | 0.7164 |

Table 2: The results on ogbl-wikikg2 with different networks. The 'R' in Network ID indicates the reduced' attention module, operating like equation (2). Train Time refers to the practical time spent for a whole training process, and every experiment is performed on a single A100 GPU.

| Network ID | hidden-dim | attn-dim | mlp-ratio | Train Time | Test MRR | Valid MRR |
|---|---|---|---|---|---|---|
| 256-32-4 | 256 | 32 | 4 | 28.7h | 0.7042 | 0.7093 |
| 512-64-4 | 512 | 64 | 4 | 48.7h | 0.7141 | 0.7189 |
| 256-R32-4 | 256 | 32 | 4 | 27.4h | 0.7067 | 0.7085 |
| 512-R8-2 | 512 | 8 | 2 | 33.3h | 0.7129 | 0.7146 |

**Subgraph Components**   On the basis of anchors, other types of information are added to subgraphs in turn. The results in Table 3 show that the addition of both path and supplementary nodes information can make obvious improvement. The subgraphs containing all kinds of information have the best results overall.

## 4.4   RESULTS ON OGBL-WIKIKG2 & FB15K-237

We combine the advantaged strategies and modifications in section 4.3 to conduct experiments, and present some of our best results in Table 4 and 5, with the corresponding experiment settings given in Appendix C, showing the trade-off between the number of parameters and the results.

Our method achieves state-of-the-art results on ogbl-wikikg2 with a significant 3%-ish improvement. We presume that the construction of incomplete multi-hop subgraphs is a summarization and extraction of the basic concepts in a knowledge graph, and much information is already stored in subgraph structures, making the learning task easier and thus leading to better results, especially for large-scale knowledge graphs. On fb15k-237, our results are constrained by the distance-based model, for the parameters in StarGraph are optimized through the score function of TripleRE'. So we directly train the embeddings of entities and relations with TripleRE' as the conduct group. StarGraph does not achieve as significant improvement on fb15k-237 as on ogbl-wikikg2 though, the results are still comparable with its conventional KGE counterpart with less parameters, which validates that StarGraph dose work on smaller KGs as well.

As with the number of parameters, the efficiency of StarGraph is more evident on large-scale KGs, as the network itself takes up a certain amount of parameters. Comparing the #Params marked in gray, the number of embedding parameters of StarGraph on fb15k-237 is only about 30%$\approx (2304 \times 512)/(14505 \times 256)$ of the control item, but the total number of parameters is about 80%; the number

Table 3: The results on ogbl-wikikg2 with subgraphs containing different types of information. The ✓indicates the corresponding type of information to be contained.

| anchors | path | center | neighbors | Test MRR | Valid MRR |
|---|---|---|---|---|---|
| ✓ | | | | 0.7067 | 0.7085 |
| ✓ | ✓ | | | 0.7222 | 0.7373 |
| ✓ | | ✓ | | 0.7197 | 0.7235 |
| ✓ | | | ✓ | 0.7248 | 0.7307 |
| ✓ | | ✓ | ✓ | 0.7248 | 0.7319 |
| ✓ | ✓ | ✓ | ✓ | 0.7239 | 0.7403 |

Table 4: The results on ogbl-wikikg2. Results of related work are taken from the OGB Leaderboard.

| Model | #Dims | #Params | Test MRR | Valid MRR |
|---|---|---|---|---|
| TransE | 500 | 1,250,569,500 | 0.4256±0.0030 | 0.4272±0.0030 |
| RotatE | 250 | 1,250,435,750 | 0.4332±0.0025 | 0.4353±0.0028 |
| PairRE | 200 | 500,334,800 | 0.5208±0.0027 | 0.5423±0.0020 |
| AutoSF | - | 500,227,800 | 0.5458±0.0052 | 0.5510±0.0063 |
| ComplEx | 250 | 1,250,569,500 | 0.5027±0.0027 | 0.3759±0.0016 |
| TripleRE | 200 | 500,763,337 | 0.5794±0.0020 | 0.6045±0.0024 |
| TripleREv2 | 200 | 500,763,337 | 0.6045±0.0017 | 0.6117±0.0008 |
| ComplEx-RP | 50 | 250,167,400 | 0.6392±0.0045 | 0.6561±0.0070 |
| AutoSF + NodePiece | - | 6,860,602 | 0.5703±0.0035 | 0.5806±0.0047 |
| TripleREv2 + NodePiece | 200 | 7,289,002 | 0.6582±0.0020 | 0.6616±0.0018 |
| TripleREv3 + NodePiece | 200 | 36,421,802 | 0.6866±0.0014 | 0.6955±0.0008 |
| InterHT + NodePiece | 200 | 19,215,402 | 0.6779±0.0018 | 0.6893±0.0015 |
| TranS + NodePiece | 200 | 19,215,402 | 0.6882±0.0019 | 0.6988±0.0006 |
| TranS(large) + NodePiece | - | 38,430,804 | 0.6939±0.0011 | 0.7058±0.0018 |
| | 256 | 7,148,802 | 0.7222 | 0.7373 |
| StarGraph + TripleRE' | 512 | 44,819,970 | 0.7263 | 0.7407 |
| | 512/32 | 93,039,522 | 0.7290 | 0.7327 |

Table 5: The results on fb15k-237. Results of related work are taken from the corresponding papers, despite TransE being provided by Dai Quoc Nguyen et al. (2018).

| Model | #Dims | #Params | Test Hit@10 | Test MRR |
|---|---|---|---|---|
| TransE | 100 | 1,474,200 | 0.465 | 0.294 |
| RotatE | 1000 | 29,484,000 | 0.533 | 0.338 |
| PairRE | 1500 | 22,468,500 | 0.544 | 0.351 |
| GC-OTE | 400 | - | 0.550 | 0.361 |
| CoKE | 256 | $\approx 10,190,000$ | 0.549 | 0.364 |
| NodePiece + RotatE | 200 | $\approx 3,200,000$ | 0.420 | 0.256 |
| TripleRE' | 256 | 3,895,296 | 0.5429 | 0.3455 |
| | 1000 | 15,216,000 | 0.5520 | 0.3514 |
| StarGraph + TripleRE' | 512 | 3,148,290 | 0.5454 | 0.3426 |
| | 512 | 4,275,714 | 0.5475 | 0.3459 |

of of embedding parameters of StarGraph on ogbl-wikikg2 is $1\% \approx (20000 \times 256)/(2500604 \times 200)$ of the control item and the total number of parameters is only 1.4%.

## 5    CONCLUSION

In this paper, we propose StarGraph, a novel method to learn knowledge representations by generating and encoding the incomplete 2-hop subgraph for the node. The experimental results verify that our method is parameters-efficient and can obtain better or comparative results with fewer parameters on different datasets. More importantly, StarGraph achieves significant improvement of results on large-scale knowledge graphs.

Based upon the core idea of incomplete subgraph, we have proposed several strategies and modifications for the implementation. Though each strategy and modification is proved to be effective alone, it is worth further exploration on how to make them work collaboratively to the best effect and whether there are better alternatives. Additionally, beside the transductive link prediction, StarGraph is also able to perform other related tasks, such as inductive link prediction, node classification, etc. All these are treated as future work.

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

Table 6: The results on ogbl-wikikg2 with different anchor sets.

| skip-threshold | anchor-set-size | anchor-ratio | $|A|$ | Test MRR | Valid MRR |
|---|---|---|---|---|---|
| 1.0 | 20,000 | 0.008 | 20,000 | 70.88 | 71.09 |
| 0.5 | 20,000 | 0.008 | 20,000 | 70.67 | 70.85 |
| 0.2 | 20,000 | 0.008 | 20,000 | 70.65 | 70.85 |
| 1.0 | 80,000 | 0.032 | 80,000 | 71.01 | 71.30 |
| 0.5 | 80,000 | 0.032 | 80,000 | **71.12** | 71.53 |
| 0.2 | 80,000 | 0.032 | 79,157 | 71.03 | **71.66** |

Table 7: The results on fb15k-237 with different anchor sets.

| skip-threshold | anchor-set-size | anchor-ratio | $|A|$ | Test Hit@10 | Valid Hit@10 |
|---|---|---|---|---|---|
| 1.0 | 1,450 | 0.1 | 1,450 | .53.55 | 54.15 |
| 0.5 | 1,450 | 0.1 | 1,450 | 53.97 | 54.58 |
| 0.2 | 1,450 | 0.1 | 1,450 | 53.80 | 54.43 |
| 1.0 | 14,505 | 1.0 | 14,505 | 54.33 | 54.93 |
| 0.5 | 14,505 | 1.0 | 4,503 | **54.75** | 55.06 |
| 0.2 | 14,505 | 1.0 | 2,304 | 54.54 | **55.12** |

## A  EXPERIENCE ON ANCHOR SET GENERATION

During the generation of anchor set, the hyper-parameter *skip-threshold* is used to control the sparsity of anchors, the smaller value the sparser anchors; *anchor-set-size* decides the maximum number of anchors, which can also be set with *anchor-ratio*, the proportion of anchors occupying in the graph. Note that when *skip-threshold* is small, skipping too many nodes may cause the actual size of the anchor set $|A|$ to be smaller than the specified *anchor-set-size*.

The experimental results with different values of *skip-threshold* and *anchor-set-size* are displayed in Table 6 and 7, where we can summarise some experience as follows. When the *skip-threshold* is fixed, the larger $|A|$, the better results. But when the *skip-threshold* varies, the situation becomes a little tricky. Comparing the results on ogbl-wikikg2 when *skip-threshold*=20,000, we can draw the conclusion that when the anchor ratio is extremely small, a high *skip-threshold* to keep the best few known nodes as anchors should bring better results. But when the anchor set exceeds a certain size, such as 80,000 on ogbl-wikikg2 and 2,000+ on fb15k-237, a lower *skip-threshold* leads to better results with an equal or even smaller $|A|$. Anyway, a proper *skip-threshold* helps to make a better trade-off between the number of parameters and the results.

## B  EXPERIMENTS WITH DIFFERENT ENCODERS

To explore the effectiveness of self-attention network to encode graphs, we conduct the control experiment of different encoders and present the results in Table 8. MLP consists of two linear layers and a non-linear activation between them, Attention+Mean is the encoder adopted in StarGraph, and Mean omitting the attention module is just a mean-pooling operation performed over all tokens. Note that the case marked in gray is actually the same method as NodePiece, in spite of the *mlp-ratio* here is 4 instead of 2 in their experiments.

Table 8: The Test/Valid MRR on ogbl-wikikg2 with different encoders in different situations,

| Sampling | path | MLP | Mean | Attention+Mean |
|---|---|---|---|---|
| BFS | | 0.6886/0.6924 | 0.6749/0.6733 | 0.6899/0.6953 |
| balanced | | 0.7079/0.7152 | 0.6596/0.6587 | 0.7067/0.7085 |
| balanced | ✓ | 0.7094/0.7247 | 0.7101/0.7250 | 0.7222/0.7373 |

Table 9: Experiment settings for StarGraph on ogbl-wikikg2.

| $|A|$ | Sampling | Network ID | path | nbors&center | learning rate | Test MRR | Valid MRR |
|---|---|---|---|---|---|---|---|
| 20,000 | balanced | 256-R32-4 | ✓ | | 1e-4 | 0.7222 | 0.7373 |
| 80,000 | balanced | 512-R8-2 | ✓ | | 5e-5 | 0.7263 | 0.7407 |
| 20,000 | balanced | 512-R8-2 | | ✓ | 2e-4 | 0.7290 | 0.7327 |

Table 10: Experiment settings for StarGraph on fb15k-237.

| $|A|$ | Sampling | Network ID | path | nbors&center | learning rate | Test Hit@10 | Test MRR |
|---|---|---|---|---|---|---|---|
| 2,304 | balanced | 512-R8-2 | ✓ | | 5e-4 | 0.5454 | 0.3426 |
| 4,503 | balanced | 512-R8-2 | ✓ | | 5e-4 | 0.5475 | 0.3459 |

Compare results with different anchors-sampling strategies, the *balanced* sampling proposed by us achieves 1%+ improvement for both MLP and StarGraph. But when it comes to the introduction of path information, MLP does not process the subgraphs as effectively as StarGraph. By the way, it is interesting to compare the results of Mean with or without the path information. The proposed infusion way of path information, which is developed form the KG distance-based model as mentioned in section 3.2.2 , are proved to be effective for the graph structure by the 5%+ improvement.

Overall, the experimental results confirm that the self-attention network cooperates the best with our subgraph-based representation learning.

## C  IMPLEMENTATION AND HYPER-PARAMETERS

The experiment settings for StarGraph in Table 4 is given in Table 9. For each setting, the best learning rate is chosen from {5e-5, 1e-4, 2e-4} according to the experimental results.

For the results in Table 5, model TripleRE' corresponds to the conventional KGE method, the dimension of embeddings is provided as #Dim, and the best learning rate is 2e-4 in {1e-4, 2e-4, 5e-4, 1e-3}. The experiment settings for StarGraph is given in Table 10, and the learning rate is chosen from {1e-4, 2e-4, 5e-4, 1e-3}.

The selection of the feature dimension and the combination of different modifications are in fact restricted, mainly due to the video memory size. Note that adopting multiple advantaged strategies and modifications does not always result in the sum of improvements, and the best learning rates for different networks or different sizes of anchor sets are usually different. It will take a lot of work in the future to explore more of the potential for StarGraph.

