# OpenReview forum: "StarGraph: Knowledge Representation Learning based on Incomplete Two-hop Subgraph"
_ICLR.cc/2023/Conference — Submitted to ICLR 2023_

### Official Review · Reviewer_AAWP · 2022-10-18

**Confidence:** 5
**Correctness:** 3
**Technical Novelty And Significance:** 2
**Empirical Novelty And Significance:** 3
**Recommendation:** 3

**Clarity, Quality, Novelty And Reproducibility:**

## Quality
Experiments in this paper are convincing and the proposed method shows strong performance. However, some concerns about motivation and ablations should be addressed. (See Strength And Weaknesses for more details.)

## Clarity
This paper is overall well written, while the way to introduce Methodology can be improved.


## Originality
Most of the used techniques are well-known, but the authors did a good job of combining them to achieve superior performance.

## Reproducibility
Detailed experimental settings and codes are provided, so the reproducibility looks nice.

**Strength And Weaknesses:**

## Strength
1. The idea of using subgraphs around entities to generate expressive entity embeddings is reasonable.
2. The authors propose a bag of tricks to reduce the computational overload for different steps of knowledge representation learning.
3. The proposed StarGraph significantly outperforms previous SOTA models on the wiki90m-v2 dataset.
4. Detailed experimental settings and codes are provided, so the reproducibility looks nice.


## Weaknesses
1. **Motivation**
    1.1 Why we can use incomplete subgraphs is not well motivated. In the introduction, the authors use Figure 1(a) to demonstrate that complete subgraphs contain a lot of redundant information. However, is there any evidence that this information is redundant? Obviously, the completed subgraph in Figure 1(a) contains more topology information than the incompleted subgraph in Figure 1(b). Although using incomplete subgraphs may lead to competitive results, we can not tell which one is more suitable for knowledge representation learning just through a toy figure illustration.

    1.2 Why we should use an anchor-based strategy is unclear. Even if using incomplete subgraphs is necessary for large-scale knowledge graphs, we can use a sample-based method (e.g., GraphSAGE) without anchors.
2. **Related Work**
    The authors did a thorough review of related work. However, the differences between StarGraph and some existing works (e.g., NodePiece and GraphSAGE) are not fully discussed.
3. **Methodology**
    3.1 The overall pipeline is unclear. I strongly suggest the authors provide a figure illustration for the overall architecture of StarGraph.

    3.2 In Section 3.2, the authors claim that they use a single transformer block. I am wondering if it is a standard transformer block, i.e., is there a residual connection and layer norm in the used module?

    3.3 The authors highlight two-hop subgraphs in the title. I am not sure whether two-hop is necessary. Can we use one-hop or three-hop subgraphs?

    3.4 The path information is captured by TripleRE in Section 3.2.2. However, it seems that the proposed framework is irrelevant to the path encoders. Therefore, I suggest the authors try more KGE models when encoding paths.

    3.5 I am curious about how to induce Equation (8) from (7). It would be better to provide a detailed derivation.
4. **Experiments**
    4.1 It'd be better to put the main results before ablation studies.

    4.2 The Network ID in Table 2 is confusing.

    4.3 According to Table 3, the combination of anchors and neighbors already leads to the best test MRR, which demonstrates that path and center are unnecessary.

    4.4 As the authors claimed, the main contribution of this paper is a new graph encoder. Therefore, the graph encoder should be able to improve the performance of different decoders not only the proposed TripleRE', e.g., TransE, RotatE, AutoSF.




**Summary Of The Paper:**

The authors propose a new knowledge representation method named StarGraph, which serves as an encoder and aims to generate high-quality entity embeddings for downstream score functions. The proposed StarGraph mainly consists of two stages: subgraph sampling and encoding, and the authors manage to optimize the efficiency in both stages. Experiments demonstrate that StarGraph achieves a new SOTA result on a large-scale knowledge graph benchmark (wik90m-v2), while being more parameter efficient than other baseline models.

**Summary Of The Review:**

This paper proposes an effective method to achieve new SOTA results on large-scale knowledge graph completion. Although the performance is strong, some concerns about motivation and ablations should be carefully addressed before this submission can be published.

===============Update After Rebuttal===============

I have read the author's response (the authors did not upload a revised submission). However, my concerns about the motivation and methodology are not addressed. The author fails to provide sufficient evidence to explain why we must design the model as they do now. In my view, this submission looks more like a technical report than an academic paper. Thus, I am leaning to reject it.

---

> ### Author Response · Authors · 2022-11-19
> **Discussion of Weaknesses 1**
>
> Thank you very much for reading our work and giving us your valuable comments!
> Discussion of Weaknesses.
> 1.1We propose to use incomplete subgraphs because multi-hop subgraphs in large-scale graphs can easily exceed the size limitation, and an intuitive idea is to utilize partial neighborhood information instead of the complete multi-hop subgraph. Fig.1 is not used to demonstrate the complete subgraphs contain redundant information, but is to illustrate the complete subgraphs might contain redundant information. The introduction brings up the problem and gives the solution, and the experiments provide the evidence. We think it is a common understanding that the size of complete multi-hop subgraphs of large-scale KGs is too large to handle well, we propose the incomplete subgraph solution, and prove that it works.
>
> 1.2 As with GraphSAGE, the sampling during each iteration would consume much time and cannot be performed on KG with the scale as OGB. And we think that the anchor-based strategy summarize information with a few fixed nodes would build strong connections for the node representations. Or just because it is a recently proposed effective work. It meets our needs, and that’s enough for us to chose it.
>
> 2.They are different at all.
>
> 3.2 Yes, it is a standard transformer block, including residual connection and layer norm. In the new paper version, we have add such description to 3.2 first paragraph to make it more clear.
>
> 3.3 It is not necessary, but reasonable to use two-hop subgraph. As stated in the Introduction, we do not consider three-hop subgraph for two reasons. 1) “Due to the large number of nodes and edges, multi-hop subgraphs in large-scale graphs can easily exceed the size limitation, and the subgraphs generation and network calculations can both be very time-consuming.” 2) “Neighbors out of reach within two hops are not as closely related to the target node according to us, so are not taken into consideration.” The one-hop subgraph was not chosen because we intuitively thought that the two-hop subgraph would contain richer contextual information. We highlight two-hop subgraph in the title not only because we use two-hop subgraphs, but also the anchor-sampling method and the path information fusion ways are based on two-hop subgraph, the “two-hop” is method related. Nevertheless, the fact that StarGraph is based on two-hop subgraphs does not mean that “two-hop” is the only or best option. We see subgraph sampling as a promising research direction and would be interested to see more related work.
>
> 3.4 Yes, the path information fusion is irrelevant to the score function, but there are good reasons we choose TripleRE. First of all, the infusion of path information is a time-consuming operation, and we want to keep it as simple as possible. The calculation of TripleRE is much simpler than interHT and TranS, while it outperforms the remaining KGE models. Moreover, the original Transformer simply add the position embedding to the token (token+position), which can be interpreted as the TranE model (t=h+r). A step further would lead to a similar form of PairRE (t=h*r), then TripleRE (t=h*r1+r2). Based on the above considerations, we choose TripleRE that is with more capacity.

---

> ### Author Response · Authors · 2022-11-19
> **Discussion of Weaknesses 2**
>
>
> 3.5 Eq.(8) is not the same as (7), but a modified version based on (7). The u in Eq.(7) is a hyper-parameter designed based on experience, and its position is not unchangeable. We pick u from Eq.(7) and put it to a different place, creating a different formula, i.e. Eq.(8). This operation has been described in section 3.3, just two lines above Eq.(8).
>
> 4.1 Thank you for your advice, and we have considered this order. However, since we proposed modifications on different aspects, the training settings in our experiments are a little complex. It is difficult to put the main results and discussions ahead without cause confusions. In order to make the experiments more readable, we decided to stay with the current order.
>
> 4.2 In title of Tab.2, we explain the ‘R’ in Network IDs indicates the reduced attention module. Beside this, the three numbers separated by - correspond to the hidden-dim, attn-dim, and mlp-ratio respectively. We did not explicitly state this because this information is non-essential to the understanding of the paper. You can also ignore the meaning of numbers in Network IDs, as long as you know that different IDs correspond to different network structures.
>
> 4.3 In Tab.3, anchors+neighbors leads to the best test MRR, but the combination of all elements leads to the best valid MRR and best mean MRR. Beside this, the ‘path’ makes a better trade-off between model parameters and performance. The ‘center’ is not our best option, but it’s a useful experience, so is also listed as supplementary information.
>
> 4.4 The main contribution is the entire method, including the idea to represent nodes with subgraphs and the strategies to generate and encode the subgraphs, and to achieve SOTA performance with the proposed method and TripleRE’. Though not claimed in the paper, we do believe our method is able to improve the encoder performance over different KGE models, e.g., TransE, RotatE, AutoSF. Since the code will be publicly available, we would welcome everyone to try StarGraph with different KGE models .

---

> ### Author Response · Authors · 2022-12-12
> **Answers to ====Update After Rebuttal ====**
>
> Thank you very much for reading our rebuttal and giving us your valuable comments! The following are our answers.
>
> "my concerns about the motivation and methodology are not addressed" ----------
> We propose to use incomplete subgraphs because multi-hop subgraphs in large-scale graphs can easily exceed the size limitation. In large graph, a node may have tens or hundreds or thousands or even more neighbors, imagine how many nodes will be in its multi-hop subgraph, and how much time and space it'll take to calculate the embeddings of all entities based on the subgraphs. We do not say the complete subgraph shall never succeed, but there are good reasons we choose to explore methods based on incomplete subgraphs. Beside this, the motivations of the two-hop, Transformer, anchor sampling, path embedding, etc., are all given in the paper where we propose the modications. The motivations do not say we "must" do what we do, but show such modification is promising and we should give it a try.
>
> "The author fails to provide sufficient evidence to explain why we must design the model as they do now" ----------
> There is no "must". There are some problems found, which may be solved with the proposed model that did get positive results. Enough reasons, aka our motivation, have been given for why we would like to design our method this way. And the experimental results demonstrated our method to be valid. In our view, we got the job done right. And if you think we shouldn't or cannot "design the model as we do now", you should provide sufficient evidence for it.
>
> "this submission looks more like a technical report than an academic paper" ----------
> We do not understand how you define a technical report or an academic paper. An academic paper, according to us, should provide something new and helpful and inspiring. We believe it is a consensus that, "something new" does not mean nothing like existing work, but is about offering a new way of understanding problems and solving them. We proposed a new way to solve the KG embedding task by learing the embeding for each entity based on its incomplete two-hop subgraphs. Our method is a combination of several proposed modifications, none of which is suggested or implied by existing work, some modifications (e.g. using Transformer layer) may be easier to think of, but most modifications and as a whole method are undoubtedly novel. All in all, we found a problem existing in previous methods, proposed a new solution, and demonstrated its effectiveness, hoping that this work will inspire other researchers to together explore more possibilities. What's more should you ask for "an academic paper"?

---

### Official Review · Reviewer_keWz · 2022-10-23

**Confidence:** 4
**Correctness:** 3
**Technical Novelty And Significance:** 2
**Empirical Novelty And Significance:** 2
**Recommendation:** 6

**Clarity, Quality, Novelty And Reproducibility:**

The paper is clear in expressing the authors’ ideas, while the proposed method is not innovative enough.

As for the reproducibility of the paper, the authors describe related parts carefully in the paper, and it is not difficult to reproduce the experimental results from the paper.

**Strength And Weaknesses:**

The strength of the paper:

In order to save time as well as keep much information from the neighbor nodes,  the authors propose a new method based on an incomplete 2-hop subgraph for knowledge representation learning.

The weakness of the paper:
1. There is an error in the last two rows of the Table. 5 obviously.
2. The proposed model is not evaluated enough on different datasets and different tasks. In the paper, the authors only evaluate the proposed method on two datasets on one task.
3. In fact, the proposed model is a variant of NodePiece and is not innovative enough.
4. In my opinion,  different types of anchors sampling methods are needed to be investigated in the ablation study.


**Summary Of The Paper:**

Considering the current KG embedding methods of conventional shallow embedding models, which ignore the contextual information,   as well as the generic graph neural networks, which are difficult to expand to a KG with a large number of nodes and are very time-consuming, the authors propose a new knowledge representation learning method based on incomplete two-hop subgraph to leverage the neighborhood information with little computational complexity.

**Summary Of The Review:**


The current version is not suitable for ICLR.

---

> ### Author Response · Authors · 2022-11-19
> **Discussion of Weaknesses**
>
> Thank you very much for reading our work and giving us your valuable comments!
> Discussion of Weaknesses.
> 1. There is actually no error in Tab.5, as which we believe you refer to the different #Params under the same #Dims? The difference is caused by the number of anchors instead of the dimensions, and the detailed settings are given is Appendix C, as we mentioned at the fist paragraph of section 4.4. To avoid such misunderstanding in the future, we add a new column to Tab.4 and 5, indicating the number of embedded nodes, i.e. the number of anchors in StarGraph case, to make the different settings more clear.
>
> 2. On one hand, more datasets and tasks would certainly make the work more convincing, and we’ll keep working on that. On the other hand, the present experimental results are sufficient to demonstrate the validity of the method, so we consider this to be a complete research work. We would like to share the current progress to provide inspiration to other researchers for further explorations.
>
> 3. In fact, NodePiece did a great work but also left many gaps. They introduced the tokenization into knowledge graphs, got obvious improvement on OGB wikikg-2, but did not give proper explanations. They adopt the anchor strategy to save parameters, without exploring the deep reason it works or its potentials. In our work, we argue that the anchor-sequence for each node implicitly contains contextual information, so the anchors should be used to connect neighbor nodes, and this idea leads to our anchor-sampling method. Moreover, considering the graph structure, the anchors should be used to construct subgraph rather than sequence, so we adopt the transformer encoder and change the position embedding to path information fusion. Comparing with NodePiece, there is a new way to understand the anchor strategy and several new methods based on the new idea and obvious improvements brought by the methods. We have made appropriate references for the part based on NodePiece, and none of our proposed methods/modifications is suggested or implied by NodePiece. We really can't agree with the statement of “a variant of NodePiece”.
>
> 4. We do not quite understand the sentence, does it mean a) we should explore more anchor sampling methods, or b) the investigation of anchor-sampling methods should be put in section 4.3 instead of Appendix, or c) there should be more discussion about the sampling methods? For b) and c), section 3.1.2 gives detailed description about the motivation and steps, and the first paragraph in section 4.3 discusses the effectiveness of the proposed sampling method. As for a), we found a problem existing in previous methods, proposed a new solution, and demonstrated its effectiveness, hoping that this work will inspire other researchers to together explore more possibilities.

---

### Official Review · Reviewer_1SnH · 2022-10-24

**Confidence:** 3
**Correctness:** 3
**Technical Novelty And Significance:** 3
**Empirical Novelty And Significance:** 3
**Recommendation:** 8

**Clarity, Quality, Novelty And Reproducibility:**

This paper is generally well written. However, the following two questions should be further considered:
1. The transformer  network is able to mix the information of all nodes in the subgraph, but it fails to model the structure of the subgraph. Therefore, it seems not a good choice to use transformer  to encode graphs. It is neccessary to explain how the transformer  model the graph strucutre. Why do not you not use GNN networks?
2. When predicting the missing tail entity for (h,r,?), the methods need to sample and encode subgraphs for all the candidate tail entities. It seems  time-consuming. It is better to provide an analysis of  time complexity.


**Strength And Weaknesses:**

Weakness:

1. The transformer  network is able to mix the information of all nodes in the subgraph, but it fails to model the structure of the subgraph. Therefore, it seems not a good choice to use transformer  to encode graphs. It is neccessary to explain how the transformer  model the graph strucutre. Why do not you not use GNN networks?
2. When predicting the missing tail entity for (h,r,?), the methods need to sample and encode subgraphs for all the candidate tail entities. It seems  time-consuming. It is better to provide an analysis of  time complexity.


**Summary Of The Paper:**

This paper proposed a StarGraph method to learn entity representations for large-scale knowledge graphs. The StarGraph first generates an incomplete two-hop neighborhood subgraph for each target node by picking part of the anchors and nodes within the 2-hop neighborhood. Then a transformer network is applied to obtain the entity representations.

1. The transformer  network is able to mix the information of all nodes in the subgraph, but it fails to model the structure of the subgraph. Therefore, it seems not a good choice to use transformer  to encode graphs. It is neccessary to explain how the transformer  model the graph strucutre. Why do not you not use GNN networks?
2. When predicting the missing tail entity for (h,r,?), the methods need to sample and encode subgraphs for all the candidate tail entities. It seems  time-consuming. It is better to provide an analysis of  time complexity.


**Summary Of The Review:**

This paper is generally well written and the solution is ok. If the authors can clarity my questions, I will re-evaluate the paper.

---

> ### Author Response · Authors · 2022-11-19
> **Discussion of Weaknesses**
>
> Thank you very much for reading our work and giving us your valuable comments!
> Discussion of Weaknesses.
> 1.For knowledge graph, one idea is to model it by graph structure or subgraph structure using networks like GCN, GAT, etc. But for the case of many nodes, GCN will consume a lot of resources to explain the computation with subgraphs, which are also very time consuming to build and compute. Another idea is to model the node embededding based on the simple relationship between nodes, such as transE, TripleRE. this article is based on the second idea to do, and we compare with these algorithms. For StarGraph, it is actually confusing to define the relationship between the target node and the anchor node, the relationship between them is not direct, so at first we did not consider the GCN approach, but of course this may also get good results, we can try it. As you said, the traditional Transformer can fuse the contextual information of the nodes in one dimension and does not incorporate the structure of the graph, but of course we can actually solve this problem by constructing relative position encoding to add this structural information in. Here we do not model the structure in a complex way, but add the more fuzzy embedding corresponding to different nodes corresponding to different paths in the transformer, and the effect has some improvement.
>
> 2. During training, we randomly sample a fixed size of negative entity (128 in our experiments) instead for all the negative ones for each triplet. During testing, for the OGB wikikg2 dataset, it specifies the positive and negative examples that need to be validated, each triplet (head, relation, tail) gets 1,000 negative entities (500 for head and 500 for tail). So there is no need to compare the positive entity to all the rest, and the prediction does not take much time. For fb15k-237, we compare the positive head and tail with all the entities, and it’s time-consuming. So we at first cache all the node embedding, as long as the embedding of each node is calculated, when comparing between nodes, only a function (which can be a simple similarity calculation, or a linear layer, etc.) is needed to calculate the relevance of the nodes. The process does not take much time, so we omit analysis of time complexity.

---

### Official Review · Reviewer_ZZ4V · 2022-10-27

**Confidence:** 4
**Correctness:** 4
**Technical Novelty And Significance:** 2
**Empirical Novelty And Significance:** 2
**Recommendation:** 3

**Clarity, Quality, Novelty And Reproducibility:**

Well writing, Complete content, The proposed method works but lacks instructions.

**Strength And Weaknesses:**


Strength:

Experiment on the ogbl-wikikg2 link prediction (Table 4) get SOTA results.

Weakness:

The proposed method in the paper has limitations. The proposed method is to select better nodes in graph, which is a widely used idea [1][2][3]. Such an approach used on the knowledge graph lacks background and motivation. Also, as a KG-independent method, experiments on big graph are missing.

Treating the knowledge graph as a simple directed graph ignores the semantic properties of the knowledge graph, and such work is not conducive to the development of knowledge graph research. For example, whether the selected nodes make important at the knowledge level, the experiment lacks a case study to demonstrate this.

Compared to the previous approach [4], the proposed method experiments on knowledge graph related tasks and it is difficult to show that it is generalizable.

[1]. Alsentzer E, Finlayson S, Li M, et al. Subgraph neural networks[J]. Advances in Neural Information Processing Systems, 2020, 33: 8017-8029.

[2]. Wang X, Ji H, Shi C, et al. Heterogeneous graph attention network[C]//The world wide web conference. 2019: 2022-2032.

[3]. Veličković P, Cucurull G, Casanova A, et al. Graph attention networks[J]. arXiv preprint arXiv:1710.10903, 2017.

[4]. Galkin M, Wu J, Denis E, et al. Nodepiece: Compositional and parameter-efficient representations of large knowledge graphs[J]. arXiv preprint arXiv:2106.12144, 2021.

**Summary Of The Paper:**

The paper argue that Conventional method can not utilize  neighborhood representation effectively for large-scale knowledge graph. The paper propose STARGRAPH, a node selection strategy, to find the superior node in two-hop subgraph. The results show that the proposed method get SOTA results in large-scale knowledge graph link prediction.

**Summary Of The Review:**

The proposed method is effective, but further reflection on KG is missing.

---

> ### Author Response · Authors · 2022-11-19
> **Discussion of Weaknesses.**
>
> Thank you very much for spending time reviewing our paper and providing constructive suggestions!
>
> 1. Methods with limitations can still be a nice one for problem-solving because there are no methods that have no limitations. The performances of a model heavily depend on the quality of data, which has been widely known for decades. All data processing methods, like data mining and augmentation, have limitations. Of course, we humans are born with limitations.
>
> 2. For StarGraph, instead of finding better nodes, the reason it gained good performances on the OGB and other datasets is that it learned to ignore redundancy information for efficiently modeling the relationships between nodes on large knowledge graph data and reducing the computational cost.
> For example, you can locate an unfamiliar place by describing the geographical location relationships between this place and some well-known places. And still, you need to choose carefully which well-known places you want to list for a quick understanding of the location. If you read our article carefully, you should be able to feel that we attach great importance to the requirement of robustness in the process of designing algorithms, precisely because considering only a small number of anchors increase the robustness of the model. Researchers familiar with graph networks should be well aware that networks such as GCN are very easy to overfit when the number of layers becomes large.
> Resnet is a special case of HighwayNet, but the motivation, to solve the problem of deep neural network gradient back propagation, is different. Resnet makes very important progress, but some fancy network does not. StarGraph has achieved good results by simplifying large knowledge graphs and using a small number of anchors as the representation of nodes, but based the idea of node embedding as modeling knowledge graphs, which is different from previous work. Of course, it has its limitation that the effect decreases on small knowledge graphs because more information is omitted, but we should not deny the progress.
>
> 3. It's not quite clear what the so-called big graph is. The OGB dataset is already publicly available as a very large knowledge graph, and we achieved a very large lead over second place on it using StarGrah.
>
> 4. For knowledge graphs, one idea is to model them by graph structure or subgraph structure, using networks like GCN, GAT, etc. But for the case with many nodes, GCN consumes a lot of resources to explain the computation of subgraphs, and the construction and computation of subgraphs are also very time-consuming. Another idea is to model node embedding based on simple relationships between nodes, such as transE, TripleRE. This paper is based on the second idea, and we compare it with these algorithms. It is precisely based on the so-called "semantic properties of the knowledge graph", the "big graph", where the relationship between nodes is not very deterministic, and the relationship between A and B can be either r1 or r2. The addition of these semantic information, on one hand, makes the model confused, on the other hand, involves the problem of data leakage. NLP model learns semantic information based on the large-scale pre-training data. When we talk about the semantic features without pre-training, the model can be simply seen as the addition of the relationship node graph, in essence, or simple directed graph.

---

### Decision · Program_Chairs · 2023-01-20

**Decision:**

Reject

**Justification For Why Not Higher Score:**

The main reason for not rating the work higher is that it seems somewhat incremental relative to the previously published work on NodePiece and does not constitute a substantial step change to be published at ICLR.

**Justification For Why Not Lower Score:**

N/A

**Metareview: Summary, Strengths And Weaknesses:**

The paper presents a new approach for representation learning in knowledge graphs. It claims that other approaches do not sufficiently represent the information in the neighborhood of knowledge graph nodes. In order to achieve this, they sample the 2-hop-neighborhood and for target nodes. They claim that the method scales better than other methods which do consider the neighborhood of nodes.

The following strengths and weaknesses were identified by the reviewers.

Strenghts:
- Reasonable method for embedding the graph neighborhood
- Positive evaluation results

Weaknesses:
- Work is inspired bye.g.  NodePiece published at ICLR 2022 and the delta to this and other methods considered to be relatively small
- Some further evaluations on larger graphs could be conducted to support the argument of scalability raised in the paper
- Some uncertainty about the methodology pointed out in parts of the reviews